# Oceanographical Context of the First Bloom of the Silicoflagellate *Octactis speculum* (Ehrenberg) Recorded to Cause Salmon Mortality in a Galician Ria: Was This Bloom a Rare Event in the Iberian Coast?

**DOI:** 10.3390/toxins15070435

**Published:** 2023-07-02

**Authors:** Ricardo Prego, Rafael Carballeira, Yolanda Pazos, Roberto Bao

**Affiliations:** 1Instituto de Investigaciones Marinas (IIM-CSIC), Eduardo Cabello, 6, 36208 Vigo, Spain; prego@iim.csic.es; 2Cavanilles Institute for Biodiversity and Evolutionary Biology (ICBiBE), University of Valencia, José Beltrán 2, 46980 Paterna, Spain; 3Instituto Tecnolóxico para o Control do Medio Mariño de Galicia (INTECMAR), Peirao de Vilaxoán, s/n, 36611 Vilagarcía de Arousa, Spain; ypazos@intecmar.gal; 4Grupo de Investigación en Cambio Ambiental (GRICA), Centro Interdisciplinar de Química e Bioloxía (CICA), Universidade da Coruña, Rúa As Carballeiras, 15071 A Coruña, Spain

**Keywords:** harmful algal blooms, HAB, fish mortality, marine phytoplankton, hydrography, nutrient salts, salmon aquaculture

## Abstract

Harmful algal blooms are one of the leading causes of mortality in salmon aquaculture, with significant economic consequences. From 15 to 31 October 1996, a bloom of the skeletonized form of *Octactis speculum* (Ehrenberg) F.H. Chang, J.M. Grieve & J.E. Sutherland was detected in the small Merexo inlet (1.7 km^2^ area), located on the southern shore of the Ria of Muxía (Galicia, NW Spain). The *O. speculum* population inside the inlet (data period: 1992–1996) seldom exceeded 4·10^3^ cell·L^−1^. However, its concentration reached 2·10^5^ cell·L^−1^ during the bloom, coinciding with a decrease in light penetration from 5 to 2 m deep, as measured using a Secchi disk. Although similar concentrations were reported during late October 1992, this was the first time that a bloom was associated with caged salmon (*Salmo salar*, Linnaeus 1758) mortality in the Galician coastal waters. This mortality was not associated with anoxia in the water column, but with fish gill irritations and mucus segregation due to gill clogging. Excess nitrate and silicate, the latter being essential for skeleton formation, were measured in the inlet during the bloom, with phosphate acting as the limiting nutrient (high negative correlation). Blooms of *O. speculum* occurred in autumn–winter, when water was retained within the inlet under meteorological conditions of southwest winds (which prompted downwelling conditions) and clear skies. A review of the oceanographic database of the Galician rias showed that massive *O. speculum* proliferations are also commonplace in other rias with similar environmental conditions, such as the Ria of Ares-Betanzos, and can therefore constitute a threat for the development of salmon aquaculture on this coast.

## 1. Introduction

Salmon aquaculture is of great economic importance, as this species is one of the most widely consumed fishes throughout the world. However, gross economic losses are reported annually as a consequence of massive fish mortality in coastal salmon farms due to the effect of harmful algal blooms (HABs) [1,2,3,4]. In fact, the incidence of harmful algae blooms on salmon farms has been extensively documented in the past 50 years in Northern Europe [2,3,4], but it is still poorly known in Southern Europe.

*Octactis speculum* (Ehrenberg) F.H. Chang, J.M. Grieve & J.E. Sutherland (Phyllum Ochrophyta, Class Dictyochophyceae) is a common silicoflagellate species in Galician coastal waters (NW Iberian Peninsula). It was first reported in the Ria of Vigo in 1951, cited as *Distephanus speculum* (Ehrenberg) Haeckel [5]. Since then, this cosmopolitan photosynthetic and marine silicoflagellate has appeared as a regular component of phytoplankton communities in the region [6,7]. *O. speculum* is typically found in cold waters—from 11° to 15 °C—where it shows its optimal growth [8]. It occurs in two distinct flagellated stages, including a naked form, first described by Van Valkenburg and Norris [9], and a skeletonized form with an internal basket-shaped silica skeleton, usually showing six spines (cell sizes: 20–40 µm). Blooms of the naked stage were reported at optimal 15–25 salinity values, whereas the skeleton-bearing stage occurs at somewhat higher salinities [8]. In 1983, a bloom of the naked form was identified in Danish waters [10], and from that year on, several other cases of massive proliferations were reported on the European coasts [11], such as in the Baltic Sea [12] and the Trieste Gulf [13].

*Octactis speculum* is considered a harmful species for marine fauna [14,15,16,17]. Harmful algal blooms (HABs) of this species have been reported since the early 1970s, causing massive mortalities of farmed salmon in Canada [18,19], caged rainbow trout in Denmark [10], and caged sea trout in France [20]. HABs of *O. speculum* usually involve the skeletonized form [21]. The skeletons provoke irritation and mucus secretion in fish gills, lowering gill exchange potential and eventually leading to death, as observed in trout [20], farmed salmon [22,23], and occasionally, wild salmon [19]. In contrast, others [8] also suggested overnight dissolved oxygen deficiency due to excessive respiration during blooms of the naked form of *O. speculum* as the triggering mechanism of mortality, which was proposed as the most likely explanation for the fish kill events that occurred in Denmark in 1983 [10] and 2004 [24]. The toxicity of the naked form has also been disputed [25].

During the second fortnight of October 1996, a mortality of 4925 caged salmon (*Salmo salar* Linnaeus 1758) occurred in a fish farm inside a small inlet in the Galician Ria of Muxía. The search for the causes of salmon mortality in this farm led us to unexpectedly investigate the presence of *O. speculum* in the Merexo inlet with the following two objectives: (i) to describe the first HAB of *O. speculum* reported to cause a massive fish kill in the Iberian Atlantic coast in relation to the available meteorological, hydrographical, and nutrient concentration data, and (ii) to give insight into whether the blooms of this silicoflagellate are unusual or common in the Galician Rias.

## 2. Study Area

The Ria of Muxía, one of the eighteen Galician Rias (Figure 1), is located in the middle of the Galician coast (NW Iberian Peninsula). The ria, an incised flooded valley whose innermost part acts as an estuary, has a 17 km^2^ surface area with a maximum depth of 27 m at its mouth, which is open to the Atlantic Ocean along an east–west axis. The ria receives its main freshwater contribution from the east end of the Grande River (average annual flow of 8.4 m^3^·s^−1^; [26]) and it is under the influence of the seasonal upwelling region of Cape Finisterre [27]. This upwelling system results from the prevailing northerly winds occurring in the region from March to October. In contrast, southerly winds prevail during the rest of the year. The climate in the Ria of Muxía is classified as Cfb (temperate oceanic climate), according to the Köppen–Geiger scheme [28]. Rain falls mostly during the autumn and winter, with an annual average rainfall of 73.5 mm. Northern winds prevail from March to October, while winds come mostly from the South during the rest of the year.

The small Merexo inlet (Figure 1) is located in the southern shore of the ria, occupying a surface area of 1.7 km^2^ with a maximum of 12 m at its mouth and containing a seawater volume of 11.5 Hm^3^. The inlet head falls into two watercourses including the Negro River, with a flow of 0.2–1.0 m^3^·s^−1^, and the Moraime stream, which contributes about 0.1 m^3^·s^−1^. A salmon farm with floating circular cages was established inside the inlet in the 1990s and ceased its activity in 1997 (Station 2 in Figure 1). This type of farming was first established in Galicia in the Ria of Ortigueira in 1976, but has not been present since 2005 [29,30,31], with the exception of some experimental cages.

## 3. Results and Discussion

### 3.1. Octactis speculum Bloom and Salmon Mortality in the Merexo Inlet

A bloom of *O. speculum* was detected in the Merexo inlet (Ria of Muxía) from 21 to 26 October 1996, reaching concentrations higher than 40,000 cell·L^−1^ (Figure 2). The bloom event occurred under prevailing soft southwesterly breezes concurring with slightly overcast skies from 11 October to 7 November, which is an unusual meteorological situation in the Galician coast. These wind conditions are known to cause seawater retention inside the Galician Rias [32,33] and are often accompanied by red tides in the early autumn [34]. During the aforementioned event, the temperature in the seawater column around the salmon cages in the Merexo inlet ranged from 14.5 to 15.0 °C, while the salinity range was 35.4–35.5 (Figure 3); the sea was calm, and the Negro River flow did not exceed 1 m^3^·s^−1^.

The *O. speculum* bloom in the Merexo inlet, therefore, lasted just a few days. The overall meteorological and hydrographical conditions, in terms of the light period (11 h a day with a clear sky), seawater temperature, and salinity, were optimal for the bloom [9,35]. Moreover, seawater retention occurring at the same time that the silicoflagellates were proliferating led to their accumulation inside the inlet.

The meteorological conditions changed in November. The wind shifted to the northeast during the first fortnight, which not only allowed for water exchange with the ocean, but also allowed for its intensification by an out-of-season upwelling event [36,37]. Precipitation increased the Negro River flow to 3 m^3^·s^−1^ and, consequently, the salinity of the surface seawater in the Ria of Muxía decreased to 27.0, close to the mouth of the Negro River (Station 4, Figure 1), 32.7 in Station 3 (Figure 1), 32.0 around the salmon cages, and 32.5 in the mouth of the Merexo inlet (Station 1, Figure 1), i.e., the positive estuarine circulation returned. Despite the fact that both the seawater temperature (≈13.5 °C) and nutrient availability (Figure 3) remained favorable, the concentrations of *O. speculum* dropped to around 1100 cell·L^−1^. In the following days, both the hydrodynamical conditions, which prompted phytoplankton dispersion, and cloudy skies, which hindered photosynthetic activity, drove the algal bloom to its end.

The Merexo lnlet has the following two main nutrient sources: a fluvial one, which comprises the Negro River and the Moraime stream (Figure 1), and an oceanic one, constituted by the surrounding Ria of Muxía. The nutrient salt concentrations in the waters of the Negro River during the bloom were 36 µM for nitrate, 1.0 µM for phosphate, and 127 µM for dissolved silicate. After the November bloom, those concentrations decreased slightly (34 µM of nitrate, 0.8 µM of phosphate, and 112 µM of dissolved silicate), while the river flow increased. The levels of nutrient salt in the Moraime stream remained around 40 µM for nitrate, 1.8 µM for phosphate, and 127 for µM dissolved silicate. Both river courses showed well-oxygenated waters, with a 100% dissolved oxygen saturation. The nutrient concentrations in the surface waters in the area surrounding the salmon cages (Station 2 in Figure 1) after the bloom (November) were 18 µM for nitrate, 0.6 µM for phosphate, and 19 µM for dissolved silicate (Figure 3). These values are approximately twice the concentrations recorded at a 3–9 m depth, where the salinities were higher than in the surface waters (Figure 3) due to the lack of river influence. Overall, the recorded nutrient concentrations were comparable to those of other well-preserved Galician rias, such as those measured during an annual cycle in the neighboring Ria of Laxe [38]. In contrast, the nutrient salt concentrations during the silicoflagellate bloom (Figure 3) were much lower, with 5 µM of nitrate, <0.05 µM of phosphate, and 2.7–4.3 µM of dissolved silicate. The *O. speculum* cell density in the Merexo inlet shows a significant positive correlation with temperature (*p* = 0.995, *p*-value < 0.05) and a negative correlation with nitrate (*p* = −0.901, *p*-value < 0.01) and, especially, with phosphate levels (*p* = −0.978, *p*-value < 0.05). There is no significant correlation with silicate concentration (*p* = −0.719, *p*-value > 0.05). The following observed statistical correlations fit well into the downwelling scenario when the bloom materialized: (a) *O. speculum* growth took place under the warm temperatures associated with SW winds, which triggered the downwelling conditions [32,33,34,35] (positive correlation with temperature); (b) cell proliferation induced phosphate and nitrate consumption, reducing the concentration of these nutrients in the water column (negative correlations with phosphate and nitrate); and (c) the granitic nature of the Galician river basins favors an excess of silicate availability for silicoflagellate growth (no significant correlation with silicate). Phosphate was found to be the limiting nutrient for this species in the inlet, as shown by the extremely low concentrations of this nutrient recorded at depth, and by the 300:1 and 40:1 nitrate/phosphate ratios recorded during and after the bloom, respectively. Although nitrate has always been considered the limiting nutrient in the Galician coast [39,40], phosphate took this role in the Merexo inlet, despite the river contributions. Most of this river-derived nutrient input could have been consumed by the bloom, since the inlet is isolated from offshore water. Nitrate replacing phosphate as the limiting nutrient was also pointed out to be responsible for a bloom of the naked form of *O. speculum* in Denmark [12]. The remaining nutrient, dissolved silicate, was found to be in excess during that HAB. In the seawater from the Merexo inlet, the nitrate–silicate relationship varied very little during (1.9:1) versus after (1.6:1) the bloom, despite the fact that the synthesis of the silicoflagellate skeleton requires silicate. Excess silicate input from the river can therefore very likely explain the favorable conditions for a bloom of the skeletonized form of *O. speculum*, instead of the naked form, in the Merexo inlet, although others [12] suggested that the growth of the naked form is apparently independent from the dissolved silicate availability.

The environmental conditions occurring at the Merexo inlet during the bloom triggered this massive growth of *O. speculum* (Figure 2). A concentration of 4350 cell·L^−1^ was recorded on 14 October, increasing to 40,890 cell·L^−1^ on 21 October, and reaching its peak on the following day, with 213,000 cell·L^−1^ (Figure 2). Thereupon, the bloom decreased to 83,500 cell·L^−1^ on 26 October, declining to the background level of 4350 cell·L^−1^ four days later, on 30 October. The seawater was turbid, and the light penetration decreased during the bloom from 5 to 2–3 m deep, as measured using the Secchi disk. Contrary to the observations made in Douarnenez Bay in France [20], where the cell concentrations reached values as high as 1,300,000 cell·L^−1^, brown-colored waters were not observed in the Merexo inlet. However, both locations recorded a high mortality of farmed salmon, which showed gill irritation and mucus segregation caused by gill clogging associated with the presence of *O. speculum*. A total of 4925 salmon died in the Merexo inlet.

Death by anoxia or oxygen depletion due to intensified respiration during silicoflagellate blooms was suggested as a cause for massive fish mortality in caged sea trout [20] and farmed salmon [22]. In fish-farm waters in the Merexo inlet (Station 2, Figure 1), the oxygen concentrations measured in October were higher than 230 µM O_2_, reaching 95% dissolved oxygen saturation at the surface and oversaturation (>120%) near the bottom (Figure 3) due to photosynthetic activity. The opposite pattern occurred one month later, once the bloom disappeared. The surface layer recorded 100% saturation, which is typical of equilibrium conditions at the seawater–atmosphere interface in the ria during the winter, while depletion was recorded near the bottom (80% saturation, Figure 3), which was probably caused by the remineralization of detritus from salmon culture and phytoplankton sedimentation [34]. It is therefore unlikely that anoxia or oxygen depletion was the cause of salmon mortality at the Merexo inlet. Alternatively, the extremely high *O. speculum* siliceous cell densities could be responsible for mechanically damaging salmon gills, as recorded in France [20] and Scotland, where others [23] documented the abrasion of delicate gill lamellae by silicoflagellate skeletons, resulting in farmed fish mortality. Damage caused by blooms of the naked form of *O. speculum* have also resulted in edema and detachment of epithelial cells in gills, as revealed by the examination of histological samples [19].

### 3.2. Octactis speculum Blooms in the Galician Rias

The aforementioned *O. speculum* bloom event in the Merexo inlet in October 1996 was not unique at this site. Although the monitoring program in the Ria of Muxía only ran from 1992 to 1996, the data analyses for these five years showed that the *O. speculum* blooms were recurrent (Table 1). The highest concentration of this silicoflagellate in the Merexo inlet was recorded in October 1992, which, at the count of 245,250 cell·L^−1^, makes this bloom the largest ever reported in the entire Galician coast. This event was followed in importance by the October 1996 event discussed in the previous section, which was outstanding because of the massive caged salmon mortality that was caused, constituting, to the extent of our knowledge, the only case ever reported in the Iberian coasts. Hence, this episode was not included in previous reviews on silicoflagellate HAB occurrences in Europe [24,25]. Other minor blooms, both in terms of concentration and duration, occurred in the Merexo inlet during 1993 (Table 1), suggesting that the natural hydrographical conditions that promote these massive growths [8,35] are common at that place.

Blooms of *O. speculum* unrelated to fish mortality are not uncommon in other Galician rias. Within the 5-year monitored period (1992–1996), high concentrations of the skeletonized form of this silicoflagellate were recorded in the Ria of Ares-Betanzos (Table 1), whereas in the remaining Atlantic Rias (Figure 1), some very local single-day proliferations (<30,000 cell·L^−1^, except for the A Creba area in Muros; Table 1) were recorded in the rias of Muros, Arousa, and Pontevedra. Besides the Merexo inlet bloom, other remarkable events occurred during March 1996 in the Ria of Ares-Betanzos (Table 1), as indicated by the records from the four control stations located in this ria (Figure 1). As in the case of the Merexo Inlet, a similar phosphate depletion occurred, from 0.5–0.9 µM to <0.1 µM phosphate, while the nitrate concentrations were >1 µM, and the dissolved silicate values were between 1 and 8 µM (M.D. Doval, unpublished data). This observation further suggests that phosphate is not only the limiting nutrient for blooms of the naked form [10], but also of the skeletonized form of *O. speculum*.

In early spring 1996 (March 18–April 1), the *O. speculum* concentrations in the Ria of Ares-Betanzos exceeded 100,000 cell·L^−1^, reaching values as high as 207,060 cell·L^−1^ in the Sada inlet. Moreover, the silicoflagellate concentrations during blooms in this inlet ranged from 52,320 to 191,840 cell·L^−1^ during 15–22 June 1992 and were as high as 125,280 cell·L^−1^ in November 1993 (Table 1). Although the seawater temperature data are unavailable for this ria during the aforementioned period, the available annual cycle information for the neighboring Ria of Coruña indicates temperatures usually lower than 15 °C in spring, as well as during and immediately after the summer upwelling pulses [41]. The July 1992 bloom is noteworthy, since it occurred outside of the usual autumn or spring seasons, suggesting that summer downwelling conditions (e.g., Ria of Vigo, Barton et al. [42]; Ria of Pontevedra, Cruz et al. [43]) can also favor silicoflagellate blooms, similarly to what occurs for red tides [33,44]. On the other hand, the fact that the *O. speculum* bloom in the Ria of Ares-Betanzos affected the entire ria suggests the hypothesis that the same could be true for the Ria of Muxía in autumn 1996, but that the bloom was only detected in the Merexo inlet as it is the single monitored sector of the ria.

The rolling topography, moderate temperatures, and wave-sheltered inlets of the Galician rias make them potentially appropriate for salmon culture, but our results show that they are also auspicious for the recurrent occurrence of a massive proliferation of silicoflagellates, such as *O. speculum*, a new HAB risk that was not previously explored in the region.

## 4. Conclusions

We report the first ever recorded bloom of *O. speculum* resulting in massive caged salmon mortality on the NW Iberian coast in 1996. Contrary to the results found elsewhere, the bloom did not induce oxygen depletion conducive to anoxia in the water column and, therefore, this was not the cause of fish mortality. In contrast, gill irritation and clogging generated by the massive accumulation of skeletonized cells was the most likely cause of mortality.

The bloom concurred with unusual autumn–winter weather conditions in this coast. These conditions entailed dominant southerly breezes and mostly clear skies from 11 October to 7 November, which led to water retention within the inlet and, therefore, interrupted its ordinary estuarine circulation. Under this downwelling scenario, excess nitrate and silicate created favorable conditions for the bloom within the inlet. Silicoflagellate growth was only constrained by phosphate availability, which became the limiting nutrient according to the 300:1 nitrate/phosphate ratio and the statistical negative correlation between the cell and phosphate concentrations. Blooms of the skeletonized form of this species can therefore be expected at unusually high concentrations of nitrate and silicate.

The review of the available oceanographic database on the Galician rias revealed several events with very high concentrations in other rias in the 1992–1996 period, which suggest that blooms of *O. speculum* are a frequent but unexplored phenomenon in this coast. Successful salmon farming is therefore potentially at risk in the Galician rias.

## 5. Material and Methods

### 5.1. Oceanographical and Meteorological Data

Oceanographical data were obtained during (30 October 1996) and after (26 November 1996) the bloom in one section of the Ria (in the four stations shown in Figure 1), with depth increasing from the innermost area of Station 4 (4 m deep) to the outermost area of Station 1 (13 m deep). Vertical temperature and salinity profiles were obtained using a Sea-Bird 19 CTD, and light penetration was estimated by means of a Secchi disk. Additionally, in October and November 1996, both meteorological (wind, rainfall, and cloudiness) and hydrographical data (surface seawater temperature and turbidity estimated from Secchi disk depths) were obtained in the Merexo inlet on a daily basis by local biologists from the Instituto Tecnológico para el Control del Medio Marino de Galicia, (INTECMAR). Moreover, meteorological conditions in the Galician coast were compiled for the period of 1992–1996 from the Agencia Española de Meteorología [45] and from Tutiempo.net [46].

Seawater was sampled at depths of 0, 3, 6, 9, and 12 m, depending on the maximum depth of each station, by means of 1.7 L Niskin oceanographic bottles that were hand thrown from a small boat, used as CTD probes. Sample aliquots for dissolved oxygen (DO) determination were separated into 140 mL Winkler flasks, and DO was immediately fixed following the Winkler method, and samples were stored in darkness until their analysis within 24 h. Dissolved oxygen concentration (μM) and percentage of oxygen saturation (%) were determined following the Winkler method [47] using a 702-MS Titrino titrator (Methron). Additional sample aliquots for nutrient salt determination were separated into 50 mL plastic bottles and preserved at 4 °C in a refrigerator. Nitrate, phosphate, and silicate were quantified the following day by means of auto-analytical techniques using three separate lines in a Technicon AAII system [48] in an IIM-CSIC lab. Nutrient salt concentrations are expressed in μM units.

### 5.2. Phytoplankton Abundance Data

Concentration data of *O. speculum* in the Merexo inlet were compiled from the INTECMAR phytoplankton database. This consists of integrative sampling data of the seawater column following the procedure described in [49]. All samples were preserved in a solution of Lugol. Cell concentrations (expressed as cell·L^−1^) were estimated following the technique described by Utermöhl [50] using a Nikon Diaphot inverted microscope. The analyzed data comprised 161 days of monitoring between 1992 and 1997, when the Merexo station ceased activity. Extra counts in the inlet, outside the regular INTECMAR monitoring scheme, were also performed in 1996 before (2 October 1996 and 14 October 1996), during (21 October 1996, 22 October 1996 and 26 October 1996), and after (28 October 1996, 30 October 1996, 5 November 1996 and 26 November 1996) the bloom event, following the same methodological procedures. Moreover, additional *O. speculum* concentration data from 42 stations belonging to the INTECMAR station network, located in the five main Galician Rias, were also used for this study. The stations were distributed from North to South as follows: 4 in the Ria of Ares-Betanzos, 7 in the Ria of Muros, 11 in the Ria of Arousa, 11 in the Ria of Pontevedra, and 9 in the Ria of Vigo (Appendix A). This amount of information provided a total of 3244 data points corresponding to integrated water column concentrations of *O. speculum* from the 1992–1996 time period.

The relationship between the environmental dataset of variables and cell density of *O. speculum* in the Merexo inlet was explored using Pearson correlation analysis (P), and by applying Student’s t-test to determine the level of statistical significance of correlations between parameters. The statistical analyses were performed using XLSTAT [51].

## Figures and Tables

**Figure 1 toxins-15-00435-f001:**
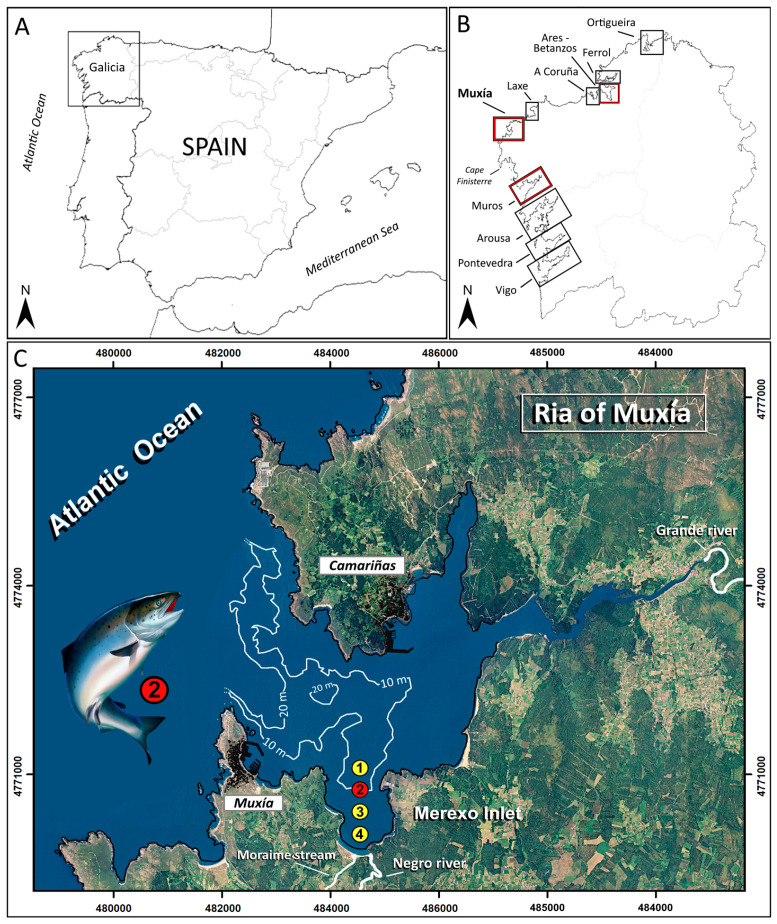
(**A**) Map of the study area (Galicia, NW Spain). (**B**) Location of the Galician rias mentioned in the text (red boxes indicate rias with *Octatis speculum* blooms reaching >40,000 cell·L^−1^). (**C**) Merexo inlet, located on the southern shore of the Ria of Muxía. The four sampling stations are shown, with Station 2 located inside the salmon farm. Rivers and streams mentioned in the text are indicated.

**Figure 2 toxins-15-00435-f002:**
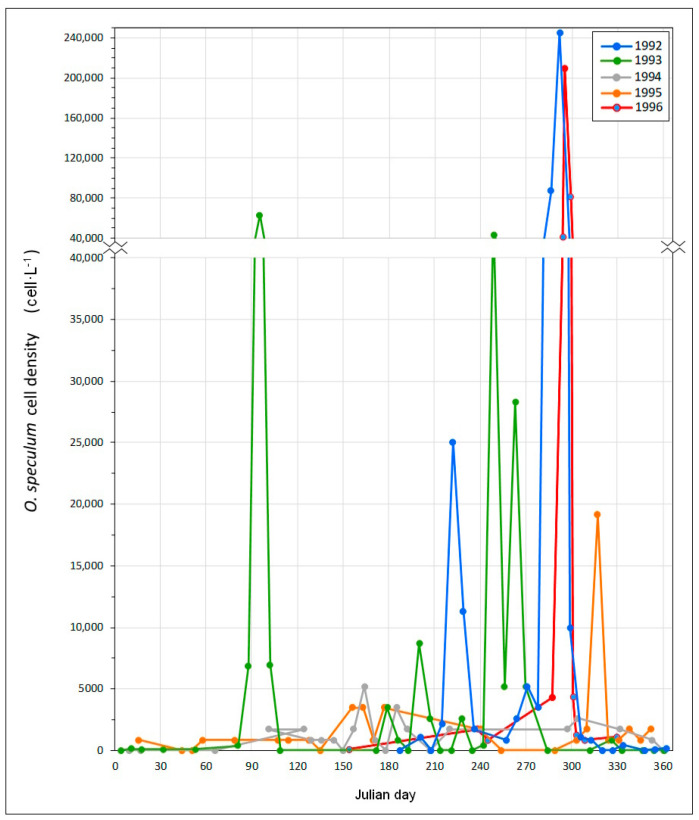
*Octactis speculum* concentrations (cell·L^−1^) in the Merexo inlet from 1992 to 1996.

**Figure 3 toxins-15-00435-f003:**
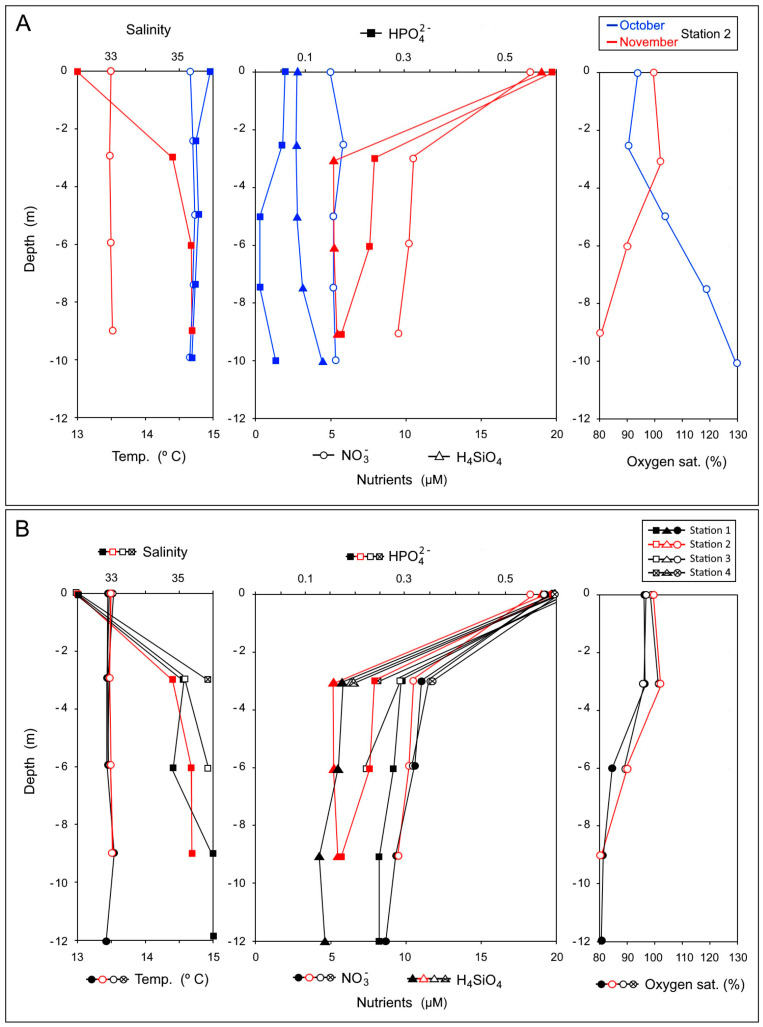
Oceanographical parameters at the four sampling stations in the Merexo inlet (Ria de Muxía, Galicia) during and after the bloom (see Figure 1), with Station 2 located inside the salmon farm. (**A**) Salinity and temperature; (**B**) nutrient salts (i.e., phosphate, nitrate, and dissolved silicates) and oxygen profiles.

**Table 1 toxins-15-00435-t001:** Concentrations of *Octactis speculum* higher than 40,000 cell·L^−1^ recorded in the seawater column of the Galician rias from 1992 to 1996 (Ria of Ares-Betanzos, four stations named Lorbé, Arnela, Sada, and Ares; Ria of Muros, A Creba).

Ria	Zone	Concentration	Date (Year)	Date (Day — Month)
Muxía	Merexo	87,000	1992	13 October
		245,250		19 October
		63,220	1993	5 April
		43,500		6 September
		40,890	1996	21 October
		209,670		22 October
		80,910		26 October
Ares-Betanzos	Amela	50,460	1992	15 July
		59,160		22 July
	Sada	52,320		15 June
		78,480		22 June
	Ares	191,840		15 June
		78,480		22 June
	Lorbé	125,280	1993	2 November
	Amela	92,220		2 November
	Sada	67,860		2 November
	Ares	52,320		5 April
		40,020		2 November
	Sada	50,460	1995	2 May
	Lorbé	146,160	1996	25 March
	Amela	175,740		18 March
		46,980		1 April
	Sada	207,060		18 March
		102,660		25 March
		41,420		1 April
	Ares	140,940		25 March
		45,240		1 April
Muros	A Creba	40,020	1992	16 June

## Data Availability

Not applicable.

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
