# Peer review of "Oceanographical Context of the First Bloom of the Silicoflagellate Octactis speculum (Ehrenberg) Recorded to Cause Salmon Mortality in a Galician Ria: Was This Bloom a Rare Event in the Iberian Coast?"

_toxins, 2023, doi:10.3390/toxins15070435_

Round 1

Reviewer 1 Report

Comments on toxins-2441337 entitled “First bloom of the silicoflagellate Octactis speculum (Ehrenberg) recorded to cause salmon mortality in a Galician ria and its oceanographical context. Was this bloom a rare event in the Iberian coast?”

A study presented in this manuscript is devoted to actual problem dealing with harmful blooming of silicoflagellate species Octactis speculum (Ehrenberg) F.H. Chang, J.M. Grieve & J.E. Sutherland, which caused of mortality in salmon aquaculture with significant economic consequences. The results are interesting, well-presented and has been carried out using standart methods. But I have some questions dealing with this manuscript.

1) Has there been any research into harmful blooming of Octactis speculum since 1996 in a Galician ria given its importance to salmon aquaculture?

2) Is the effect of phosphate on the abundance of Octactis speculum statistically significant? Has a statistical analysis been carried out?

3) Lines 37-38. The authors should indicate the taxonomic position of Octactis speculum. According to the database AlgaeBase (https://www.algaebase.org/) this species belongs to class Dictyochophyceae and phylum Ochrophyta.

4) Line 189. “4.1. Octactis speculum blooms in the Galician Rias” should be corrected to “4.2. Octactis speculum blooms in the Galician Rias”.

Author Response

REVIEWER 1 Comments

A study presented in this manuscript is devoted to actual problem dealing with harmful blooming of silicoflagellate species Octactis speculum (Ehrenberg) F.H. Chang, J.M. Grieve & J.E. Sutherland, which caused of mortality in salmon aquaculture with significant economic consequences. The results are interesting, well-presented and has been carried out using standart methods. But I have some questions dealing with this manuscript.

We thank the reviewer for his/her complementary comments to our work. In the following lines we address some issues posed by him/her and hope that we have accomplished now her/his requirements to see our manuscript published.

1)           Has there been any research into harmful blooming of Octactis speculum since 1996 in a Galician ria given its importance to salmon aquaculture?

We performed two literature surveys (on June 20, 2023) in the SCOPUS database, on one hand, using the keywords “silicoflagellate” and “salmon” and, on the other, using the keywords “salmon” and “Galicia”. The searches did not provide other similar studies from the region.

2)           Is the effect of phosphate on the abundance of Octactis speculum statistically significant? Has a statistical analysis been carried out?

We have now performed a set of correlation analyses between O. speculum abundances versus temperature and nutrient salts concentrations (i. e., nitrate, phosphate, and silicate) using the Merexo inlet dataset. We have therefore added the following contents in the text:

In the Abstract:

Lines 24 to 26: Excess nitrate and silicate, the latter being essential for skeleton formation, were measured in the inlet during the bloom, with phosphate acting as the limiting nutrient (high negative correlation).

In the Results and Discussion section:

Lines 166 to 177: O. speculum cell density in the Merexo inlet shows a significant positive correlation with temperature (P = 0.995, p-value < 0.05) and a negative correlation with nitrate (P = - 0.901, p-value < 0.01) and, especially, with phosphate levels (P = - 0.978, p-value < 0.05). There is no significant correlation with silicate concentration (P = -0.719, p-value > 0.05). The observed statistical correlations fit well into the downwelling scenario when the bloom materialized: a) O. speculum growth took place under the warm temperatures associated to SW winds which triggered the downwelling conditions [32-35](positive correlation with temperature), b) cell proliferation induced phosphate and nitrate consumption, reducing the concentration of these nutrients in the water column (negative correlations with phosphate and nitrate), and

  1. c) the granitic nature of the Galician river basins favors in-excess silicate availability for silicoflagellate growth (no significant correlation with silicate).

In the Conclusions section:

Lines 292 to 295: Silicoflagellate growth was only constrained by phosphate availability, which became the limiting nutrient according to the 300:1 nitrate/phosphate ratio and the statistical negative correlation between cell and phosphate concentrations.

In the Materials and Methods section:

Lines 346 to 349: The relationship between the environmental dataset of variables and cell density of O. speculum in the Merexo inlet has been explored using Pearson correlation analysis (P), applying a Student’s t-test to determine the level of statistical significance of correlations between parameters. The statistical analyses were performed using XLSTAT [51].

3)           Lines 37-38. The authors should indicate the taxonomic position of Octactis speculum. According to the database AlgaeBase (https://www.algaebase.org/) this species belongs to class Dictyochophyceae and phylum Ochrophyta

We have now extended on the taxonomic position of O. speculum. Former lines 37-38 have been rewritten as:

Lines 44-45: Octactis speculum (Ehrenberg) F.H. Chang, J.M. Grieve & J.E. Sutherland (Phyllum Ochrophyta, Class Dictyochophyceae) is a common silicoflagellate species in Galician coastal waters (NW Iberian Peninsula).

4)           Line 189. “4.1. Octactis speculum blooms in the Galician Rias” should be corrected to “4.2. Octactis speculum blooms in the Galician Rias”.

Section number corrected.

Reviewer 2 Report

the authors need to revise the paper:

1- The abstract just a verbiage and did not mention any results of the nutrients salts or causes of the phenomenon.

2- In the study area the description of the study area not clear to the general reader, please explain it further.

3- No statistical processes were conducted to identify the causes of this phenomenon.

4- Research language needs careful review.

5- Nutrient salts do not mean any limiting factor for each.

6- You mentioned in materials and methods that the concentrations of Octactis speculum was expressed as cells mlL-1, while in Table 1 the concentrations were expressed as cell L-1. Please correct unit.

7- In the materials and methods , it was reported that sampling took place on October 30, 1996 and after November 26, 1996. Then you mentioned the period of sampling 1992-1996 with references, so how is that??

8- In the materials and methods you must mention the unit of DO, nitrate, phosphate and silicate.

9- In the materials and methods phytoplankton abundance data you mentioned the number 3244, what do you mean?

Research language needs careful review

Author Response

Reviewer 2 Comments

1-           The abstract just a verbiage and did not mention any results of the nutrients salts or causes of the phenomenon.

Following the reviewer’s overall comments on English usage throughout the text, the entire manuscript has now been revised by an American native English speaker. All changes are indicated in the marked version of the manuscript.

As for the causes of O. speculum blooms, we think that they were already stated in our first version of the Abstract (see the highlighted texts in bold below), and that extending on them would unnecessarily enlarge its length. However, we have now included downwelling in the coast as a condition for the blooms:

Lines 23-31: This mortality was not associated with anoxia in the water column, but with fish gill irritations and mucus segregation due to gill clogging. Excess nitrate and silicate, the latter being essential for skeleton formation, were measured in the inlet during the bloom, with phosphate acting as the limiting nutrient (high negative correlation). Blooms of O. speculum occurred in autumn-winter, when water was retained within the inlet under meteorological conditions of southwest winds (which prompted downwelling conditions) and clear skies. A review of the oceanographic data-base of the Galician rias showed that massive O. speculum proliferations are also commonplace in other rias showing similar environmental conditions, such as the Ria of Ares-Betanzos, and can therefore constitute a threat for the development of salmon aquaculture on this coast.

2-           In the study area the description of the study area not clear to the general reader, please explain it further.

We agree with the reviewer that the description was very probably too short. We have now extended and rearranged it for greater clarity as follows:

Lines 80-100: The Ria of Muxía, one of the eighteen Galician Rias (Figure 1), is located in the middle of the Galician coast (NW Iberian Peninsula). The ria, an incised flooded valley whose innermost part acts as an estuary, has a 17 km2 surface area with a maximum depth of 27 m at its mouth, which is open to the Atlantic Ocean along an east-west axis. The ria receives its main fresh-water contribution from the east end of the Grande River (average annual flow of 8.4 m3·s-1; [26]) and it is under the influence of the seasonal upwelling region of Cape Finisterre [27]. This upwelling system results from the prevailing northerly winds occurring in the region from March to October. In contrast, southerly winds prevail during the rest of the year. The climate in the Ria of Muxía is classified as Cfb (temperate oceanic climate), according to the Köppen-Geiger scheme [28]. Rain falls mostly during the autumn and winter, with an annual average rainfall of 73.5 mm. Northern winds prevail from March to October, while winds come mostly from the South during the rest of the year.

The small Merexo Inlet (Figure 1) is located in the southern shore of the ria, occupying a surface area of 1.7 km2 with maximum 12 m at its mouth and containing a seawater volume of 11.5 Hm3. The inlet head falls into two watercourses: the Negro River, with a flow of 0.2-1.0 m3·s-1, and the Moraime stream, which contributes about 0.1 m3·s-1. A salmon farm with floating circular cages was established inside the inlet in the 1990s and ceased its activity in 1997 (Station 2 in Figure 1). This type of farming was first established in Galicia in the Ria of Ortigueira in 1976, but has not been present since 2005 [29-31], with the exception of some experimental cages

3-           No statistical processes were conducted to identify the causes of this phenomenon.

Statistical tests performed. See above our response to reviewer’s 1 request #2 for further details.

4-           Research language needs careful review.

English language usage has been reviewed by an American native English speaker. All corrections have been highlighted in the marked version of the manuscript.

5-           Nutrient salts do not mean any limiting factor for each.

We think that both the elemental ratios data and the new statistical tests performed give now a much clearer view about nutrients as limiting factors for the development of the blooms. Please see the corresponding texts in our response to reviewer’s 1 request #2.

6-           You mentioned in materials and methods that the concentrations of Octactis speculum was expressed as cells mL-1, while in Table 1 the concentrations were expressed as cell L-1. Please correct unit.

The referee is right. We have now corrected the units to cell·L-1 throughout the text.

7-           In the materials and methods, it was reported that sampling took place on October 30, 1996 and after November 26, 1996. Then you mentioned the period of sampling 1992-1996 with references, so how is that??.

We agree with the reviewer that these lines were quite confusing. We have therefore completely rewritten the first paragraph of section 6.2 as follows:

Lines 332-347: Concentration data of O. speculum in the Merexo inlet was compiled from the INTECMAR phytoplankton database. This consists of integrative sampling data of the seawater column following the procedure described in [49]. All samples were preserved in a solution of Lugol. Cell concentrations (expressed as cell·L-1) were estimated following the technique described by Utermöhl [50] using a Nikon Diaphot inverted microscope. The analyzed data comprised 161 days of monitoring between 1992 and 1997, when the Merexo station ceased activity. Extra counts in the inlet, outside the regular INTECMAR monitoring scheme, were also performed in 1996 before (October 2, 14), during (October 21, 22, 26) and after (October 28, 30 and November 5, 26) the bloom event, following the same methodological procedures. Moreover, additional O. speculum concentration data from 42 stations belonging to the INTECMAR station network, located in the five main Galician Rias, was also used for this study. The stations were distributed from North to South as follows: 4 in the Ria of Ares-Betanzos, 7 in the Ria of Muros, 11 in the Ria of Arousa, 11 in the Ria of Pontevedra and 9 in the Ria of Vigo (Figure S1). This amount of information provided a total of 3,244 data points corresponding to integrated water column concentrations of O. speculum from the

1992-1996 time period.

8-           In the materials and methods you must mention the unit of DO, nitrate, phosphate and silicate.

We have now included the units for DO, nitrate, phosphate and silicate (μM) in section 6.1. Dissolved oxygen saturation is also expressed as percentage (%).

9-           In the materials and methods phytoplankton abundance data you mentioned the number 3244, what do you mean?

See the re-written paragraph in our response to reviewer’s request #7.

Round 2

Reviewer 2 Report

thank you for revising and fulfilled what was required